# Statins in Children with Neurofibromatosis Type 1: A Systematic Review of Randomized Controlled Trials

**DOI:** 10.3390/children10091556

**Published:** 2023-09-15

**Authors:** Aris P. Agouridis, Nikoletta Palli, Vasiliki-Eirini Karagiorga, Afroditi Konsoula, Lamprini Markaki, Nikolaos Spernovasilis, Constantinos Tsioutis

**Affiliations:** 1School of Medicine, European University Cyprus, 2404 Nicosia, Cyprus; nikolettapalli@gmail.com (N.P.); k.tsioutis@euc.ac.cy (C.T.); 2Department of Internal Medicine, German Oncology Center, 4108 Limassol, Cyprus; 3Department of Psychiatry, Icahn School of Medicine at Mount Sinai, New York, NY 10029, USA; valiakarag@hotmail.com; 4Department of Pediatrics, General Hospital of Sitia, 72300 Sitia, Greece; aphroditekonsoula@gmail.com; 5“Iliaktida” Pediatric & Adolescents Medical Center, 4001 Limassol, Cyprus; markaki_liana20@yahoo.com; 6Department of Infectious Diseases, German Oncology Center, 4108 Limassol, Cyprus; nikspe@hotmail.com

**Keywords:** statins, neurofibromatosis type 1, NF1, children, pediatric population, systematic review

## Abstract

Background: Statins, apart from their plasma-cholesterol-lowering ability, exert several pleiotropic effects, making them a potential treatment for other diseases. Animal studies have showed that statins, through the inhibition of 3-hydroxy-3-methylglutaryl coenzyme A reductase, can affect the Ras/MAPK pathway, thus providing impetus to examine the efficacy of statins in the pediatric population with neurofibromatosis type 1 (NF1). We aimed to systematically address all relevant evidence of statin treatment in children with NF1. Methods: We searched PubMed and Cochrane Library resources up to 2 June 2023 for randomized controlled trials (RCTs) written in English and evaluating statins versus placebo in children with NF1 (PROSPERO registration number: CRD42023439424). Results: Seven RCTs were suitable to be included in this qualitative synthesis, with a total participation of 336 children with NF1. The duration of the studies ranged from 12 to 52 weeks. The mean age of the pediatric population was 10.9 years old. Three studies investigated the role of simvastatin, while four studies examined lovastatin. According to our analysis, neither simvastatin nor lovastatin improved cognitive function, full-scale intelligence, school performance, attention problems, or internalizing behavioral problems when compared with placebo in children with NF1. Statins were well tolerated in all included RCTs. Conclusion: Although safe, current evidence demonstrates that statins exert no beneficial effect in cognitive function and behavioral problems in children with NF1.

## 1. Introduction

Statins, apart from lowering serum cholesterol, exert several pleiotropic effects, some of which include neuroprotective actions, antioxidant activity, and anti-inflammatory and immunomodulatory properties, as well as thrombogenicity reduction [1,2]. The above-mentioned effects make statins a potential regimen for several other diseases [3,4,5]. Notably, statins are also recommended as first-line pharmacological option in childhood dyslipidemia (from 6–10 years old), starting at lower doses, exhibiting a positive and well-studied safety profile [6].

Neurofibromatosis type 1 (NF1), a common genetic disorder that causes learning disabilities and attention deficits is the most prevalent neurocutaneous condition. NF1 is inherited autosomal-dominantly with an incidence rate of approximately 1:2600 to 1:3000 births [7]. It is caused by pathogenic mutations in the *NF1* gene, which is located on chromosome 17q11.2 [8]. If the *NF1* gene is defective, then a genetic disease with multisystemic involvement and an increased risk of malignancies can develop [9].

Regarding clinical distinguishable manifestations of NF1, these include the so-called café au lait skin lesions, axillary and inguinal freckling, peripheral neurofibromas, and Lisch nodules [9]. Children with NF1 might present with a variety of symptoms, such as skin lesions, vision issues, seizures, learning challenges, growth retardation, endocrine abnormalities, and unexplained hypertension, among others [9].

Cognitive dysfunction is the most prevalent consequence affecting the quality of life of children with NF1 [10]. Specific learning disorders were observed in children with NF1 who were seen to have difficulty with reading, spelling, and mathematics [10]. The neuropsychological profile of NF1 is marked by abnormalities in perceptual abilities, executive functioning, and attention [10]. The expressive and receptive language abilities of NF1 children are considerably affected as well [10]. Also, the intelligence quotient (IQ) score is 5 to 10 points lower in NF1 children in comparison with the general population [10,11].

The social and learning deficits in NF1 have been associated with the loss of neurofibromin, a negative regulator of the rat sarcoma viral oncogene homologue (Ras), which causes disinhibition of the Ras/MAPK (mitogen-activated protein kinase) pathway [11]. Myelin formation and axonal integrity can also be affected by the upregulation of the Ras pathway [12]. Statins, 3-hydroxy-3-methylglutaryl coenzyme A (HMG-CoA) reductase inhibitors, cause downregulation of this Ras activation [13].

In a study performed by Li et al. [13], lovastatin improved learning and attention deficits in an NF1 mouse model by decreasing the enhanced brain p21Ras/MAPK activity [13]. In practical terms, lovastatin normalized Ras activity, repaired synaptic plasticity abnormalities, and corrected learning and attention deficits in the NF1 mouse model [13]. In addition, the inhibition of HMG-CoA reductase by statins could reverse cognitive impairments in a mouse model of NF1 disease [14].

Moreover, experiments in NF1 mouse hippocampus models showed that lovastatin altered the expression of a large number of genes, including those disturbed by NF1 mutations, suggesting lovastatin as a potent modifier of the molecular pathways that cause NF1 learning deficits [15]. In this context, several trials attempted to investigate if the above-mentioned pathway could be applied in clinical practice.

In the current review, the aim was to systematically discuss all the relevant information on statin administration in children with NF1 and to investigate whether statins exhibit favorable effects in this population.

## 2. Materials and Methods

### 2.1. Study Design

A qualitative synthesis including randomized controlled trials (RCTs) was performed focusing on the effect of statins in children with NF1 to identify if there is any beneficial outcome on cognitive function and behavior in the pediatric population.

### 2.2. Search Strategy

This systematic qualitative synthesis was registered on PROSPERO (ID number: CRD42023439424) and was carried out following the Preferred Reporting Items for Systematic Reviews and Meta-Analyses (PRISMA) recommendations [16]. An extensive bibliographic search of PubMed and Cochrane library resources was performed until 2 June 2023, using the following keywords: (neurofibromatosis type 1 OR neurofibromatosis type I OR NF1) AND (statin). In order to detect any relevant missed articles, the reference lists of qualifying articles were manually examined, as well. Only papers conducted in the pediatric population and written in English were included in the analysis.

### 2.3. Eligibility Criteria

Suitable studies for our qualitative synthesis were RCTs conducted in pediatric populations (age < 18 years old) that examined the administration of statins (atorvastatin, fluvastatin, lovastatin, pitavastatin, pravastatin, rosuvastatin, and simvastatin) versus placebo in NF1. We excluded articles focusing on adult population (>18 years old), any non-randomized studies (case–controls, cohort, and cross-sectional studies), as well as manuscripts written in a non-English language. The eligibility criteria are summarized in Table 1.

### 2.4. Data Extraction

Two authors (A.P.A. and N.P.) separately scrutinized the records to determine which studies met the inclusion criteria. Through the proceedings, any controversy between the above-mentioned reviewers was sorted out by unanimity. Because of the study’s design, approval by the National Bioethics Committee and patients’ informed permission was not necessary. Data extraction was performed using the following domains: first author, publication year, the country where the trial was conducted, number of participants, age of the participating population, study design, study duration, statin therapy, comparator, and the outcome.

### 2.5. Assessment of Risk of Bias

A risk of bias assessment was performed for each included study to establish the transparency of the systematic review results and findings, using version 2 of the Cochrane risk of bias tool for randomized trials (RoB2) [17]. The evaluation was separated into a series of domains through which bias might be introduced into trials. Each domain included a number of ‘signaling’ questions, which aimed to reach judgments focusing on the risk of bias. According to specific answers to the above signaling questions, an agendum generated a proposed judgment of “Low” risk of bias, “High” risk of bias, or “Some concerns”.

## 3. Results

### 3.1. Study Selection

Figure 1 synopsizes the outcomes of the extensive bibliographic search in a PRISMA flowchart. The initial search identified 33 publications through PubMed and 11 through Cochrane Library. We also identified one record through the reference lists. Of these, eight publications were excluded because they were duplicates between the databases. Moreover, after further evaluation of the titles and abstracts, 19 additional records were excluded as well. The full manuscripts of the remaining 18 studies were further examined for possible qualification. Studies to be incorporated in this analysis had to fulfil predetermined criteria according to the Population, Intervention, Comparison, Outcomes, and Study (PICOS) approach (Table 1). After further reviewing the full manuscripts, 11 more studies did not meet the inclusion criteria and were excluded (Table 2). After final exclusion, seven RCTs matched the requirements to be incorporated in this systematic review.

### 3.2. Study Characteristics

Seven studies were incorporated in this systematic review and evaluated 336 children, with a mean follow up period of 21 weeks [29,30,31,32,33,34,35]. Two studies [34,35] were sub-analyses of an included RCT [32], but with different outcomes. The mean age of children on statins in six out of the seven studies was 10.9 years old, while the mean age of children on placebo was 10.9 years as well. The study by Bearden et al. [31] was not used in the mean age calculation as it provided the median range of the children’s age (10–17 years old). The male to female ratio was similar in six out of the seven studies. In the seven RCTs, only two statins were investigated: simvastatin (three RCTs) and lovastatin (four RCTs). The studied population suffered from NF1. One study included NF1 patients with autism [33]. Another study used 44 patients aged 10–50 years old, but we provide data of the 14 children aged 10–17 years old that were included in the analysis [31].

The neuropsychological, neurophysiological, and neuroradiological tests that were used are, in brief, the following: the Rey CFT (complex figure test), the cancellation test (assesses attention, speed), a prism adaptation task, mean ADC value (apparent diffusion coefficient) of the brain, the Stroop color word test, the block design test and object assembly test of the Wechsler Intelligence Scale for Children–Revised, the Beery developmental test of visual–motor integration, and the judgment of line orientation task. Moreover, other measures include the CBCL attention problems (child behavior checklist, parent-rated and internalizing behavioral problems), full-scale intelligence (WISC-III-NL), teacher-reported school performance (teacher report form), parent-reported psychosocial quality of life (child health questionnaire-parent form 50 (CHQ-PF50)), patient-reported internalizing behavioral problems (youth self-report (YSR) form), fine motor coordination (grooved pegboard test), Digit Cancellation (attention/inhibitory control), and HVLT test (Hopkins Verbal Learning Test). Furthermore, other tests include ADHD (attention deficit/hyperactivity disorder), BRIEF GEC (Behavior Rating Inventory of Executive Function Global Executive Composite), COWAT (Controlled Oral Word Association Test), CPTII (Continuous Performance Test Second Edition), DT (Divided Attention), PAL (Paired Associated Learning), SOC (Stockings of Cambridge), SST (Stop Signal Task), SWM (Spatial Working Memory), ABC (Parent-rated Aberrant Behavior Checklist), CGI-S (Clinical Global Impression Scale), parent-rated Conners questionnaire for overactivity symptoms, and the Arena Maze task (assesses spatial learning).

The RCTs were multicentered and were carried out in several places around the world. These places include the USA (one RCT), USA and Australia (three RCTs), the Netherlands (one RCT), the Netherlands and Belgium (one RCT), and the UK (one RCT). The detailed characteristics of the qualified RCTs are synopsized in Table 3.

### 3.3. Outcomes of the Included Studies

The outcomes can be seen in detail in Table 3. Based on our analysis, three RCTs evaluated the role of simvastatin, suggesting a lack of beneficial effect on cognitive functioning, full-scale intelligence, school performance, attention problems, and internalizing behavioral problems. Additionally, four RCTs investigated lovastatin, showing no effect on internalizing or externalizing problems, as well as the thought, attention, or social problems in the pediatric population. The study by Bearden et al. used 44 patients aged 10–50 years old. Although beneficial effects of statins were seen in adults, no significant outcome was noted in the 14 children aged 10–17 years old [31].

### 3.4. Quality Appraisal

The bias risk was examined with the use of the RoB-2 tool (Figure 2 and Figure 3). Out of the seven studies assessed, only one study was judged to have “high” bias risk and two were judged to have “some concerns”, while the remaining four trials were judged to have a “low” bias risk.

## 4. Discussion

This systematic review marks the first comprehensive exploration of the potential link between statin administration and NF1 in the pediatric population. Our findings suggested no beneficial outcomes regarding the effect of statins in children with NF1, despite previous experimental, animal, and clinical observations that showed a possible beneficial effect of statins on NF1 mechanisms and manifestations.

NF1 is a polymorphic condition with variable clinical features including cognitive disorders and adaptation difficulties [36]. Children can develop, among others, low intellect, learning difficulties that diminish their academic potential, a sociocultural deficit, a lack of socialization, hyperactivity, autism spectrum disorders, or school difficulties with an attention deficit [36]. A higher incidence of autism spectrum disorder and cognitive problems in individuals with NF1 suggests that NF1 is linked with how Ras and other intracellular pathways are regulated [23,37].

In a phase I study, brain functional connectivity changes were evaluated after lovastatin treatment in seven children (aged 10–15 years old) with NF1 [21]. Administration of 20–40 mg/day of lovastatin for 12 weeks regulated functional connectivity, produced increased long-range positive resting-state functional connectivity, and produced less diffuse local resting-state functional connectivity in NF1 children [21].

Previous literature has also suggested that statins could impact the Ras pathway, synaptic plasticity, and cognitive functions [23]. A placebo-controlled, randomized double-blind trial conducted in adults (19–44 years old) with NF1 showed that lovastatin restores LTP-like plasticity, decreases intracortical inhibition, and improves phasic alertness when compared with controls [23]. More specifically, 200 mg of lovastatin or placebo was administered daily for 4 days in 11 NF1 patients and 11 healthy controls. Notably, the above study results revealed for the first time the association between the pathological RAS pathway activity, intracortical inhibition, and impaired synaptic plasticity with their rescuing by lovastatin in humans. Similar beneficial effects were seen by Bearden et al. in the NF1 adult population. In brief, lovastatin demonstrated beneficial effects on some learning and memory functions, as well as internalizing symptoms in the adult NF1 patients. However, no significant outcome was observed in the 14 children aged 10–17 years old [31]. These mechanisms underpin the potential clinical benefit of statins as a therapeutic option for NF1 patients with attention disorders [23]. In addition, several studies have reported the influence of statins on neuromuscular junctions, mast cell degranulation inhibition, and even the modulation of gene expression [38]. Animal studies further support the role of statins in stabilizing developmental processes and improving NF1-related symptoms [37]. Different mechanisms have been proposed to explain these observations. The HMG CoA-reductase inhibitor, simvastatin, lowers the increased frequency of spontaneous transmission of NF1 larvae at the neuromuscular junction, but still not to the level of controls who were administered a placebo [37]. This molecule nevertheless completely recovers the increased quantal size of NF1 mutants, with simvastatin also fully recovering their decreased quantal content [37].

Furthermore, other animal studies also suggest that lovastatin may play a role in stabilizing developmental processes and may be helpful in a mouse model of NF1 [13]. Also, research shows that genetic amplification of the Ras-ERK pathway does not worsen dyskinesia in mice caused by L-DOPA administration, but it does stop lovastatin from improving it [39]. Finally, regarding lovastatin, it could also modify the expression of numerous genes, including those affected by NF1 mutations. The results showed a genome-wide view of the molecular problems in the NF1 (+/−) hippocampus, which could contribute to elucidating the new molecular pathways that lead to learning problems in NF1 [15].

Several studies focused on deficits in synaptic plasticity and cognitive impairment, two of the most significant health issues in patients with RASopathies [40]. Recent results for lovastatin in patients with NF1 have shown an improvement in synaptic plasticity and cognition [40]. During synaptic plasticity and memory formation, several signaling pathways regulate gene transcription in crucial ways [15], such as the NF1 mutations anticipated that interfere with memory-related gene expression. Several of the genes that are affected by NF1, such as *Rabs*, *synaptotagmins*, *NMDAR1*, *CaMKII*, and *CREB1*, are important for synaptic plasticity [15]. For example, neurofibromin, as a GTPase-activating protein, which is encoded by the gene responsible for NF1, modulates the Ras/MAPK and cAMP signaling pathways [39].

In a randomized, triple-blind, placebo-controlled crossover trial, the neurochemical and physiological changes induced by lovastatin were investigated with magnetic resonance spectroscopy and transcranial magnetic stimulation (TMS) [28]. Lovastatin/placebo (60 mg/day, 3 days) was administered in 15 NF1 adult patients (aged 26–55 years old). According to the results of this study, lovastatin was able to modulate cortical inhibition in NF1 adults, as assessed using TMS cortical silent period ratios (relative: *p* = 0.027; absolute: *p* = 0.034).

It is worth mentioning the general studied beneficial effects of statins in NF1. Statins have shown promise in addressing cardiovascular issues associated with NF1, including reducing neointima formation and aneurysm formation [22,41]. More specifically, rosuvastatin significantly reduced neointima formation when compared with controls in neurofibromin-deficient myeloid cells, suggesting a potential therapeutic option for NF1 cardiovascular disease [22]. It has also been shown that oral administration of simvastatin reduced aneurysm formation in NF1 mice [41]. In addition, lovastatin attenuates retinal dysfunction in genetically engineered mouse models [26]. The combination of bisphosphonates and pravastatin reduced genomic instability in skin fibroblast cell lines from 43 individuals with NF1 [27]. Lastly, the combination of statins with other compounds like rapamycin, an mTOR (mammalian Target of Rapamycin) inhibitor, has shown synergistic anti-proliferative effects [42]. More specifically, the combination of simvastatin with rapamycin inhibited the fibroblast growth in NF1-related malignant peripheral nerve sheath tumor cell lines [42].

Despite the promising results from experimental animal studies as well as studies in adults, and phase I studies in children, our results suggest that statins do not exert beneficial effects in the NF1 pediatric population. None of the included 7 RCTs showed improvement in cognitive function and behavioral problems after statin administration. Although there are 7 available statins on the market (atorvastatin, fluvastatin, lovastatin, pitavastatin, pravastatin, rosuvastatin, and simvastatin), the studies retrieved for our systematic review examined the role of only 2 statins, simvastatin (3 studies) and lovastatin (4 studies), suggesting a lack of beneficial effect on cognitive functioning, full-scale intelligence, school performance, attention problems, as well as internalizing and externalizing behavioral problems. Hence, the link between statin use and clinical improvement in cognitive function and behavioral problems in children should be confirmed by future large-scale studies.

Clinical data have proven that the safety profile of statin administration is generally excellent, with reported adverse events being mostly related to muscle toxicity and liver enzyme elevation [43,44]. The occurrence of specific adverse effects, such as newly diagnosed diabetes mellitus, is considered less significant compared with the established cardiovascular benefits of statin intervention [45]. Regarding safety in our systematic review, no significant adverse events were reported in the studied pediatric population. In brief, musculoskeletal, gastrointestinal, skin, central nervous system, and respiratory system disorders were reported, but they were not different between the statin and placebo groups.

In a phase I study conducted in 24 children aged 10–17 years old diagnosed with NF1 and cognitive disabilities, Acosta et al. examined the safety profile of lovastatin [20]. According to their results, lovastatin 20–40 mg/day was well tolerated, with reports of minimal side effects. Briefly, the reported side effects included mainly cold symptoms, headache, muscle cramps, eye infections, cough, rash, stomachache, being excitable/fidgety, constipation, inability to focus, loss of balance, fatigue, bronchitis, shoulder “pops”, epistaxis, allergies, and loss of appetite. These side effects did not affect the study plan, as no withdrawal or extra intervention was required. Of note, no serious adverse event was noted.

The use of lipophilic statins, simvastatin and lovastatin, in the included studies can be explained by the numerous available data on the safety profile of the aforementioned statins in the pediatric population. The pharmacokinetic profile of simvastatin and lovastatin also plays an important role in why these two agents were chosen instead of other statins. It is a fact that hydrophilic HMG-CoA inhibitors such as atorvastatin, pravastatin, and fluvastatin manifest a very limited penetration of the blood–brain barrier [46]. As a consequence, the lipophilic statins were selected and preferred for the treatment of specific central nervous system biochemical deficits. Moreover, early intervention during childhood, which is considered the peak period of cognitive function development, can potentially strengthen the benefits of statin treatment.

### Limitations

This systematic review evaluated the effect of statin therapy in NF1 children. By extensively searching two databases, we believe that we detected all the relevant RCTs. However, certain limitations occurred. The small sample of the included participants could affect the strength of the studies’ outcomes. In addition, different scales and tools were applied in each included study for the evaluation of the findings, making it impossible to compare them, and eventually, we were unable to perform a meta-analysis.

## 5. Conclusions

The current systematic review found seven RCTs that assessed the impact of statin administration (simvastatin, lovastatin) on cognitive function and behavioral problems in the NF1 pediatric population. Based on our findings, statins should not be administered as a specific treatment of cognitive and behavioral problems in children with NF1, as they do not seem to provide any beneficial effects. Future research with larger studies is required to give more robust evidence with regard to the role of statins in children with NF1.

## Figures and Tables

**Figure 1 children-10-01556-f001:**
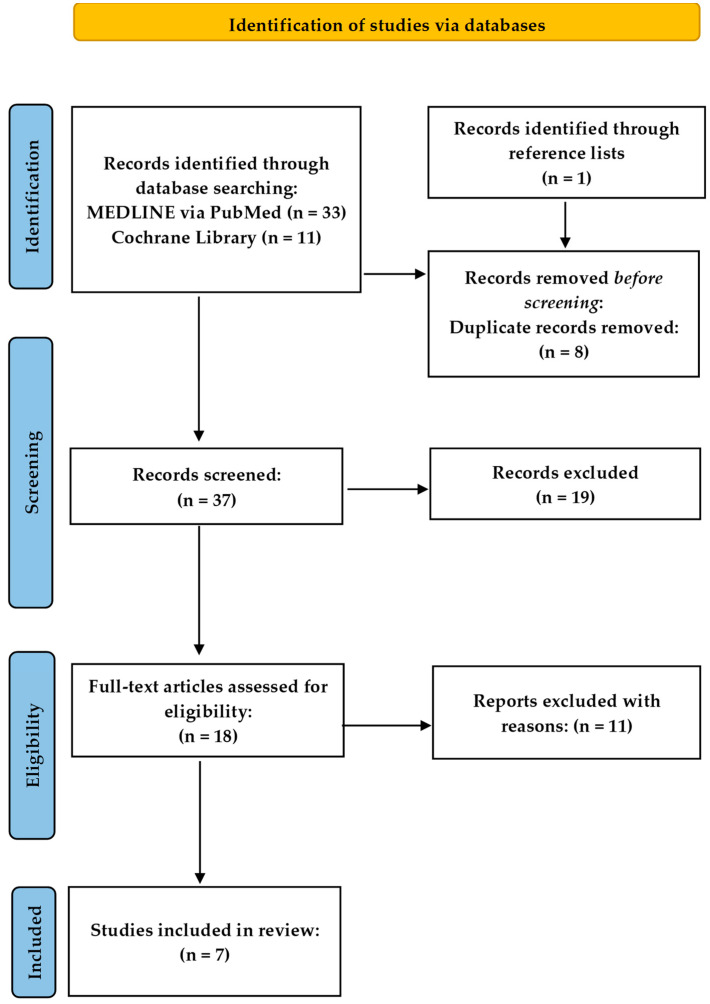
PRISMA flowchart of our systematic review.

**Figure 2 children-10-01556-f002:**
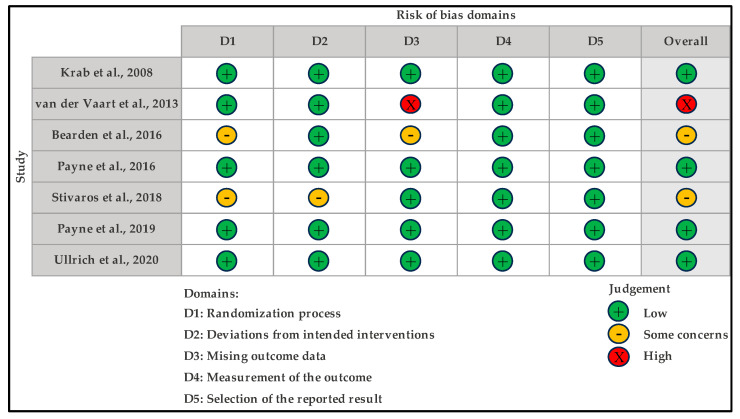
Traffic light plot for risk of bias assessment of the included studies using the revised Cochrane risk of bias tool for randomized trials (RoB-2) [29,30,31,32,33,34,35].

**Figure 3 children-10-01556-f003:**
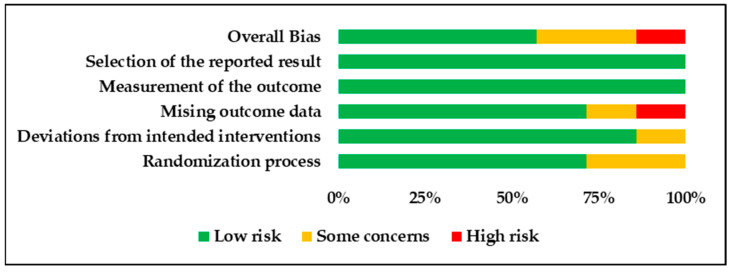
Summary plot for risk of bias assessment of the included studies using the revised Cochrane risk of bias tool for randomized trials (RoB-2).

**Table 1 children-10-01556-t001:** PICOS criteria for inclusion and exclusion of studies.

Parameter	Inclusion	Exclusion
Population	Pediatric population (<18) with neurofibromatosis type 1 (NF1)	Non-NF1 patients, age > 18
Intervention	Administration of statins	Other interventions
Comparator	Placebo or control group	Statins, other than placebo/control group
Outcomes	Effect on cognitive functioning	Other outcomes
Study design	Randomized controlled trials, published in English	All other study types, published in any other language than English

**Table 2 children-10-01556-t002:** Table of the excluded studies with reasons.

Study	Year	Reason of Exclusion
Kolanczyk [18]	2008	Study conducted in animals
Wang [19]	2011	Study conducted in animals
Acosta [20]	2011	Not an RCT
Chabernaud [21]	2012	Not an RCT
Stansfield [22]	2013	Study conducted in animals
Mainberger [23]	2013	Study conducted in adults (age: 19–44 years old)
Rosser [24]	2015	Poster, same reported results as an included study
van der Vaart [25]	2016	Observational study (sub-analysis of an included study)
Toonen [26]	2017	Study conducted in animals
Combemale [27]	2022	Study conducted in skin tissue from adults (age: 24–84 years old)
Bernardino [28]	2022	Study conducted in adults (age: 26–55 years old)

**Table 3 children-10-01556-t003:** Characteristics of the included studies.

First Author	Year	Country	Study Design	Study Duration	Population	Number of Participants (Statin/Placebo)	Age (Years)	Statin (Dosage)	Comparator	Study Outcomes
Krab [29]	2008	Netherlands	RCT	12 weeks	NF1	62 (31/31)	8–16	Simvastatin (10–40 mg/day)	Placebo	No difference in Rey CFT (delayed recall), cancellation test (speed), prism adaptation, and mean brain ADC values
van der Vaart [30]	2013	Netherlands and Belgium	RCT	12 months	NF1	84 (43/41)	8–16	Simvastatin (10–40 mg/day)	Placebo	No positive effect on cognitive functioning, full-scale intelligence, school performance, attention problems, and internalizing behavioral problems
Bearden [31]	2016	USA	RCT	14 weeks	NF1	30	(18–50)	Lovastatin (40–80 mg/day)	Placebo	Beneficial effects of lovastatin on some learning and memory functions, as well as internalizing symptoms in adult NF1 patients
						14	(10–17)	Lovastatin (20–40 mg/day)	Placebo	No effect on internalizing or externalizing problems and thought, attention, or social problems in pediatric population
Payne [32]	2016	Australia and USA	RCT	16 weeks	NF1	146 (74/72)	8–15	Lovastatin (40 mg/day)	Placebo	No effect on visuospatial learning or attention
Stivaros [33]	2018	UK	RCT	12 weeks	NF1 with Autism	30 (16/14)	8.1 ± 1.8	Simvastatin (0.5–1 mg/kg/day to a maximum of 30 mg/day)	Placebo	No significant between-group behavior effects (according to ABC, Conners, and CGI). Simvastatin was associated with increased frontal white matter and reduced grey nuclei
Payne [34]	2019	Australia and USA	Sub-analysis of RCT	16 weeks	NF1	146 (74/72)	8–15	Lovastatin (40 mg/day)	Placebo	No effect on executive functioning, visuospatial learning, attention, or behavioral problems
Ullrich [35]	2020	Australia and USA	Sub-analysis of RCT	16 weeks	NF1	29 (13/16)	8–15	Lovastatin (40 mg/day)	Placebo	No difference in spatial learning using the Arena Maze tool

Abbreviations: ABC: Aberrant Behavior Checklist; ADC: apparent diffusion coefficient; CFT: complex figure test; RCT: randomized controlled trial; NF1: neurofibromatosis type 1; CGI: Clinical Global Impression.

## Data Availability

Not applicable.

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
