# Peer review of "Statins in Children with Neurofibromatosis Type 1: A Systematic Review of Randomized Controlled Trials"

_children, 2023, doi:10.3390/children10091556_

Round 1
Reviewer 1 Report
The authors have summarized the available literature on the use of statins in children with NF1. This paper is relevant and of interest to the NF community and should be published with some significant edits.
Abstract
Line 16 - There is no controversy about the role of statins in NF1. The NF community knows clearly from the literature that they do not improve cognitive or behavioral problems so they need to edit this sentence.
Introduction
Line 48, 50 - Reference 9 - The only appropriate reference to use here would be the new NF1 diagnostic criteria. Legius E, Messiaen L, Wolkenstein P, Pancza P, Avery RA, Berman Y, Blakeley J, Babovic-Vuksanovic D, Cunha KS, Ferner R, Fisher MJ, Friedman JM, Gutmann DH, Kehrer-Sawatzki H, Korf BR, Mautner VF, Peltonen S, Rauen KA, Riccardi V, Schorry E, Stemmer-Rachamimov A, Stevenson DA, Tadini G, Ullrich NJ, Viskochil D, Wimmer K, Yohay K; International Consensus Group on Neurofibromatosis Diagnostic Criteria (I-NF-DC); Huson SM, Evans DG, Plotkin SR. Revised diagnostic criteria for neurofibromatosis type 1 and Legius syndrome: an international consensus recommendation. Genet Med. 2021 Aug;23(8):1506-1513. doi: 10.1038/s41436-021-01170-5. Epub 2021 May 19. PMID: 34012067; PMCID: PMC8354850.
-Line 59 - The evidence for the role of statins in children with NF1 is not conflicting. This sentence needs to be changed and needs to be more specific.
Discussion
- Discussion is way too long for this manuscript and should be shortened and focused to more concisely reflect the purpose of the manuscript. The manuscript unfortunately falls apart once it gets to the discussion due to the extraneous information.
- The whole discussion of statins in NF1 animal models is not well explained and does not flow. There can be some discussion of the relevant mouse and fly studies that led to human translational studies but this should be focused on neurocognitive and behavioral studies.
- The paragraph containing lines 219-226 makes no sense as written. There needs to be some explanation as to why they are discussing NF1 larvae. Why are they also discussing mast cells in this paragraph?
- Lines 247-275 - Why is there a discussion of statins in NF1 bone issues and MPNSTs? This is not the point of this paper.
- Lines 275-280 likely are relevant but are mixed in with the rest of the paragraph that discusses unrelated material
- Lines 287-296 are a good summary of the results.
Results
- Line 122 - should read "extensive literature search"
- Lines 145, 146 - the mean age should not be given as approximately. It is a solid calculated number.
- Line 151 - needs reference
- Table 3 - The last column should be the "Primary Outcome Measures." Should also be the title of the section.
- In general should discuss "statistically significant results"
Conclusion
- Line 319 - specific treatment of cognitive and behavioral problems
- Overall the English in this manuscript is relatively correct.
- Some edits need to be made with verb tenses and wording. Ex - tenses in the Methods section need to be past tense.
- Dates need to be changed to proper formatting - June 2, 2023 for example.
- Spell out the word versus. 'Vs" should not be abbreviated.
Author Response
Please find attached the authors reply.

Reviewer 2 Report
The review performed by Agouridis et al. is a well-written manuscript on statin use in NF1 pediatric individuals. I would like to compliment the authors on their Discussion section. However, I would like to address some questions regarding the Methodology:
1. Line 59: What is the rationale for the statins use in NF1 (molecular mechanisms? Protein targets?)?
2. Line 60: I suggest rephrasing the sentence: Promising results were reported in a mouse model by Li et al. [13].
3. Section 2.2 (lines 83-90): the search strategy is too vague for a systematic review. What were the combiners used? How was the search query organized considering PICOS criteria (Table 1)? Please present the Boolean operators used.
4. The "additional record" encountered disqualifies the search strategy as a systematic review.
5. Line 263. I suggest the authors mention rapamycin is not a statin and its effect on mTOR (which might also be implicated in NF1).
6. Minor point: please italicize all the genes.
Author Response
Please find attached the authors reply.

Reviewer 3 Report
Dear authors,
Children with neurofibromatosis require special attention and care, in a multidisciplinary team. It would be wonderful if new molecules could be identified to improve the quality of life and improve the cognitive dysfunctions of these patients. Unfortunately, it seems that statins do not have a beneficial effect in this regard, as your study demonstrates. Regarding the manuscript, it is well structured and apart from some typographical errors, I have nothing to complain about.
Author Response
Please find attached the authors reply.

Round 2
Reviewer 1 Report
Regarding section 3.3 and table 3 - I am not sure if the authors have verified that all the outcomes in this section and in the table are actually primary outcome measures. This should be verified and/or the title changed to perhaps "primary and secondary outcomes" or just "study outcomes."
The discussion section is still too long and not appropriately focused on the topic of statins' impact on neurocognition. This section could be written in a way to better incorporate information about the use and effect of statins in other areas of NF1 but the way it is currently written is confusing and just provides extraneous information.
I would recommend deleting lines:
-234-238
-258-266
-274-285
-286-288
English is almost perfect. Needs little to no revision.
Author Response
Please find attached the authors' reply

Reviewer 2 Report
The authors answered all my questions and performed the alterations I requested. However, I am still not convinced the study of Bearden et al. should be included in the manuscript. Since it was not encountered in the search criteria, I believe its later insertion through the reference list is not appropriate for a systematic review. My suggestion would be to exclude the study rather than alter the whole search criteria.
Author Response
Please find attached the authors' reply
